# Dispersion of Boron Nitride Nanotubes by Pluronic Triblock Copolymer in Aqueous Solution

**DOI:** 10.3390/polym11040582

**Published:** 2019-04-01

**Authors:** Sang-Woo Jeon, Shin-Hyun Kang, Jung Chul Choi, Tae-Hwan Kim

**Affiliations:** 1Department of Applied Plasma & Quantum Beam Engineering, Chonbuk National University, Jeonju 54896, Korea; ejn00013@jbnu.ac.kr; 2Department of Quantum System Engineering, Chonbuk National University, Jeonju 54896, Korea; hunny@jbnu.ac.kr; 3High Enthalpy Plasma Research Center, Chonbuk National University, Wanju 55317, Korea; happycjc@daum.net

**Keywords:** Pluronic triblock copolymer, boron nitride nanotube, hydrophobic interaction, small angle neutron scattering (SANS), dispersion

## Abstract

Boron nitride nanotubes (BNNTs) have been of interest for their excellent thermal, electrical, and mechanical properties, and they have a broad spectrum of potential applications, such as in piezoelectric materials, reinforcement of materials, and electrothermal insulation materials. For practical use of BNNTs, it is desirable to disperse them in aqueous solution, which improves convenience of handling. However, it is still difficult to make a homogenous and stable BNNT dispersion in aqueous solution, due to their strong van der Waals interactions and hydrophobic surface. To solve these problems, we used Pluronic P85 and F127, which have both hydrophilic groups and hydrophobic groups. Here, we report the wrapped structure of a Pluronic polymer-BNNT dispersion by using small-angle neutron scattering, UV–Vis spectroscopy, thermogravimetric analysis, and atomic force microscopy.

## 1. Introduction

Boron nitride nanotubes (BNNT) were theoretically predicted in 1994 [1] and first synthesized using an arc discharge in 1995 [2]; they can be imagined as a rolled-up hexagonal BN layer [3]. BNNTs have excellent mechanical [4,5,6], thermal [7] and electronic properties [8], making them highly applicable to nano- and bio-fields such as in electrically insulating thermally conductive materials [9,10,11], reinforcing composites [12,13], piezoelectric materials [14], and boron neutron capture therapy (BNCT) materials [15,16]. In spite of their extraordinary properties, BNNTs have the aggregation problem in aqueous solution (they do not disperse and exist as a bundle) due to their strong van der Waals interactions and hydrophobic surface. In order to easily utilize BNNTs in practical applications, therefore, it is desirable to homogeneously disperse them in aqueous solution. Even though a few studies on the fabrication and characterization of BNNTs have been reported, BNNT-related research, including the dispersion of BNNT in solution, is still in the very early stages. Furthermore, it is very important to disperse BNNTs, and to understand the dispersibility of BNNTs, in aqueous solution, which provides easy solvent processing in practical applications and fundamental information about the preparation of BNNT dispersions in any solution.

In this article, to disperse BNNTs in aqueous solution, we used Pluronic F127 (PEO_99_PPO_69_PEO_99_, *M*_W_ = ~12600) and Pluronic P85 (PEO_25_PPO_40_PEO_25_, *M*_W_ = ~4600) as dispersants, since these Pluronic triblock copolymers are amphiphilic [17], having both hydrophilic polyethylene oxide (PEO) groups and relatively hydrophobic polypropylene oxide (PPO) groups. Then, the hydrophobic surface of BNNTs was covered with a hydrophilic surface by wrapping the polymers on the BNNT surface, which stabilizes the BNNT dispersion in aqueous solution. The characteristics (quality, stability, and structure) of the dispersion of the Pluronic polymer-modified BNNTs were confirmed by UV–Vis spectroscopy, atomic force microscopy (AFM), thermogravimetric analysis (TGA), and small-angle neutron scattering (SANS).

## 2. Materials and Methods

### 2.1. Materials

Boron nitride nanotubes were supplied from the High-Enthalpy Plasma Research Center at Chonbuk National University (Iksan, Republic of Korea). Anhydrous ethanol was purchased from Sigma-Aldrich (Seoul, Republic of Korea). The amphiphilic triblock copolymer Pluronic F127 was purchased from Sigma-Aldrich and Pluronic P85 was obtained from BASF. H_2_O was used as deionized water. D_2_O was purchased by Cambridge Isotope Laboratories, Inc. (Tewksbury, MA, USA).

### 2.2. Sample Preparation

Before the polymer modification, as-delivered BNNTs were purified to remove impurities (amorphous boron, hexagonal BN sheets, etc.). The pristine BNNTs were calcined in air at 650 °C for 6 h (the thermal oxidation step) and dispersed in anhydrous ethanol by sonicating for 15 min. The dispersed BNNTs were then filtered through a PTFE filter (ADVANDTEC) and dried at 60 °C. The thermal oxidation step was performed in a MF-21G muffle furnace (JEIO Tech). Purified BNNTs (0.06 g) were mixed with Pluronic P85 (0.6 g) or Pluronic F127 (0.6 g) in H_2_O (30 g). To isolate the polymer-BNNTs (BNNTs with adsorbed Pluronic F127 and Pluronic P85), the polymer-BNNT mixtures were sonicated at high power (25 kHz) for 30 min followed by centrifugation at 2502× *g* for 5, 10, 15, 30, 45, and 60 min, for different test samples. For another series of samples, the centrifugation was performed for 30 min at 1118, 2008, 2502, 3007, and 4002× *g*. Thereby, F127-BNNTs and P85-BNNTs mixed in aqueous solution were separated up to 80%, and another residue sank at the bottom. The scheme of sample preparation is shown in Figure 1.

### 2.3. UV–Vis Measurements

All the UV–Vis spectra were measured by using a PerkinElmer Lamda-950 in the range of 250–800 nm. For the measurement of Pluronic polymer-BNNT samples, standard quartz cells (purchased from Hellma Analytics, Müllheim, Germany) with a 2 mm beam path length were used.

### 2.4. Atomic Force Microscopy (AFM) Measurements

The prepared polymer-BNNT dispersions were spin-coated onto silicon wafers, which had been cleaned with piranha solution. The atomic force microscopy (AFM) images were taken in tapping mode by using a Bruker AFM instrument (MultiMode 8, Bruker, Karlsruhe, Germany). To prepare the AFM samples of BNNTs, the polymer-BNNTs were deposited onto a silicon wafer and burned at 450 °C for 4 h in air to remove the Pluronic F127 and P85 adsorbed on the BNNTs.

### 2.5. Thermogravimetric Analysis (TGA) Measurements

Both the prepared F127-BNNT and P85-BNNT dispersions were freeze-dried at −43 °C for 3 days to obtain a dry powder. TGA measurements were performed by TGA Q50 (TA Instruments, New Castle, DE, USA) The powders were heated from ambient temperature to 800 °C at a heating rate of 5 °C/min under N_2_.

### 2.6. Small-Angle Neutron Scattering (SANS) Measurements

All the SANS measurements were taken by the EQ-SANS instrument at the Spallation Neutron Source (SNS) in the United States. Two different configurations (sample-to-detector distances of 2.5 and 4 m for neutrons with a minimum wavelengths of 2.5 and 10.0 Å (under 60 Hz operation), respectively) were used to cover the q range of 0.004 Å^−1^ < q < 0.5791 Å^−1^ (where q = (4π/λ) sin(θ/2) is the magnitude of the scattering vector and θ is the scattering angle). The sampled scattering was corrected for background noise, empty-cell scattering, and for the sensitivity of individual detector pixels. The corrected datasets were placed on an absolute scale using the data reduction software provided by SNS [18,19].

### 2.7. Scanning Electron Microscopy (SEM) Measurements

Scanning electron microscopy (SEM) images of the pristine BNNTs were measured using a Field Emission SEM instrument (Hitachi (SU8030), Omuta, Japan).

## 3. Results and Discussion

The prepared Pluronic triblock copolymer-BNNTs, after ultrasonication, were well-dispersed in water. To separate the bundled BNNTs in the water, which can persist even after ultrasonication, the suspension was centrifuged. The prepared BNNT dispersion in water is whitish and slightly transparent. To determine the optimal conditions for the preparation of BNNT dispersions in water, the polymer-BNNT mixtures were prepared with varying sonication times, centrifugation times, and g-forces. As the sonication time increased, the dispersibility of the polymer-BNNT mixture was increased. However, we did not observe any rod shapes in the AFM measurements (where BNNTs appear as very long rods) when the polymer-BNNT mixture was sonicated for 1 h, indicating that the BNNTs were broken by the ultrasonication after a long time. Therefore, we fixed the sonication time at 30 min (where enough rod shapes were observed in the AFM images). Depending on the centrifugation time and g-force, the dispersibilities of Pluronic F127-BNNT and P85-BNNT mixtures were assessed by visual inspection and UV–Vis measurements (Figure 2 and Figure 3). As the centrifugation time increased (at 2502× *g*), the UV–Vis absorbance of the polymer-BNNT dispersion decreased (i.e., the transparency of the dispersion was increased (Figure 2a)) and reached a minimum after a long enough time (30 min) as shown in Figure 2b–d. Therefore, we fixed the centrifugation time at 30 min, and then the centrifugation g-force in the preparation process was varied from 1108 to 4002× *g*. With increasing centrifugation g-force, the dispersion became more transparent (Figure 3a). The UV–Vis absorbance of the polymer-BNNT dispersion gradually decreased with increasing centrifugation g-forces and reached a minimum near 2500× *g* (Figure 3b–d). Therefore, we fixed the centrifugation g-force at 2502× *g* for the preparation of the polymer-BNNT dispersion in water. It should be noted that, under low centrifugation times and g-forces, Pluronic F127, with a higher molecular weight than P85, is rather effective for forming the polymer-BNNT dispersion in water, where the UV–Vis absorbance of F127-BNNT dispersions is significantly higher than that of P85-BNNT dispersions. However, under high centrifugation time and g-force, the dispersibility of the two polymer-BNNT mixtures becomes identical (their UV–Vis absorbances are eventually identical), making very stable BNNT dispersions which last for a long time (at least 1 month).

To check the content of BNNT in the solution, the polymer-BNNT dispersions prepared under the optimized conditions (both sonication and centrifugation time of 30 min and centrifugation at 2502× *g*) were freeze-dried and then TGA measurements were performed, which demonstrated that both types of polymer-BNNT powders contain ~5 wt % of BNNTs (Figure 4).

Since the BNNTs have strongly hydrophobic surfaces, they are not spontaneously dispersed in aqueous solutions. To disperse BNNTs in aqueous solution, a suitable dispersant is required, which can form a hydrophilic surface when adsorbed on the nanotubes. In addition, ultrasonication is required to break up the bundled BNNTs. When using these procedures, we expect that a Pluronic polymer-BNNT complex was formed, which makes a stable BNNT dispersion in aqueous solution. Here, it is important to understand the microstructure of the complex, which is directly related to the dispersibility of BNNTs in aqueous solution. Therefore, to understand in detail the structures of the polymer-BNNT complex in aqueous solution, SANS measurements were performed. The samples for SANS experiments were prepared in D_2_O to enhance the neutron scattering contrast between the particles and the solvent. SANS intensities of F127-BNNTs and P85-BNNTs in D_2_O (at a total concentration of 0.2 wt %) are shown in Figure 5. For the high-q region (q > 0.02 Å^−1^), SANS intensities were gradually increasing, which is typical of a Gaussian polymer coil, meaning that the solution contained some free polymer (which is not adsorbed on the BNNT surface). In the middle-q region (q ≈ 0.02 Å^−1^), SANS intensities of polymer-BNNTs in D_2_O dramatically increased proportionally to q^−a^ (where a > 1), which indicates Porod’s law scattering related to the interfacial thickness information. Due to the limited low-q range of the instrument, we did not observe the Guinier region, which would indicate some length information about the cylindrically shaped particles. Therefore, in the model-fit analysis of the SANS data, we looked for the core radius and shell thickness, and we fixed the length at 500 nm, which is much larger than the length obtained from the accessible q range in these measurements. It is natural that some of the Pluronic polymers were present in the form of Gaussian coils in the solution without being adsorbed on BNNT surfaces. Considering that a BNNT is cylinder-shaped and a lot of polymers are adsorbed on the surface, a sum of Gaussian coil and core-shell cylindrical form factors [20,21] was adopted in the SANS model-fit analysis. Since the samples were prepared in dilute suspension and interparticle interference was not observed over the whole q range, the intraparticle interference (form factor) was considered in the model-fit analysis. The sum of Gaussian coil and core-shell cylindrical form factors agreed with the SANS intensities of polymer-BNNTs in D_2_O (Figure 5) (where the core diameter (BNNT) is identical (3.0 nm) and the shell thicknesses (Pluronic polymer) were 2.57 and 2.10 nm for F127-BNNT and P85-BNNT in D_2_O, respectively). The radii of gyration for the Gaussian coils were 3.19 and 2.12 nm. The shell thickness of F127-BNNT and the radius of gyration for the Gaussian coil of F127-BNNT in D_2_O were slightly larger than those of P85-BNNT in D_2_O due to the large molecular weight of Pluronic F127. Considering that the BNNTs synthesized by the RF plasma method have varying numbers of walls and their diameters range from 2 to 10 nm, the average diameter of BNNT obtained from the model-fit analysis, which was 3.0 nm, indicates that the BNNTs are individually dispersed in aqueous solution, and their surfaces were coated with Pluronic polymer as shown in Figure 6.

To confirm the BNNT diameter in the polymer-BNNT dispersion, AFM measurements were performed. The samples for AFM measurements (bare BNNT) were deposited onto a silicon wafer (by spin-coating) and were burned at 450 °C for 4 h to remove the Pluronic polymer adsorbed on the BNNT surface. The AFM images of bare BNNT showed very long rods (where the length is estimated to be about 100–1000 nm and the diameter ranges from 2 to 7 nm) (Figure 7). Comparing the dispersed BNNTs’ structure (the diameter (a few nanometers) and length (a few micrometers)) to that of pristine BNNTs, the BNNTs were chopped during the sample preparation (upon ultrasonication and purification) and individually separated and dispersed, which is consistent with the SANS results.

## 4. Conclusions

The dispersibility of BNNTs with reformed hydrophilic surfaces was improved by using the Pluronic triblock copolymers (Pluronic F127 and Pluronic P85). From visual inspection and UV–Vis measurements, we determined optimal conditions for the preparation of polymer-BNNT dispersions, and the resulting dispersions were stable for long times. The prepared BNNT dispersions contained about 5 wt % of BNNT in water, which was confirmed by TGA measurements. SANS and AFM measurements showed that the BNNTs are individually dispersed in aqueous solution and the Pluronic polymer coated the BNNT surfaces. The adsorbed Pluronic polymers on the BNNT surfaces comprise a hydrophilic surface, and thereby the BNNTs are well-dispersed in aqueous solution. Since the BNNTs are individually separated and dispersed, the BNNT dispersions were stable for long times. This study demonstrates an easy way to disperse BNNTs with strongly hydrophobic surfaces in aqueous solution, allowing a simplification of the solvent processing of BNNTs in practical applications. Furthermore, this can give us fundamental information about the dispersion of nanoparticles with a hydrophobic surface.

## Figures and Tables

**Figure 1 polymers-11-00582-f001:**
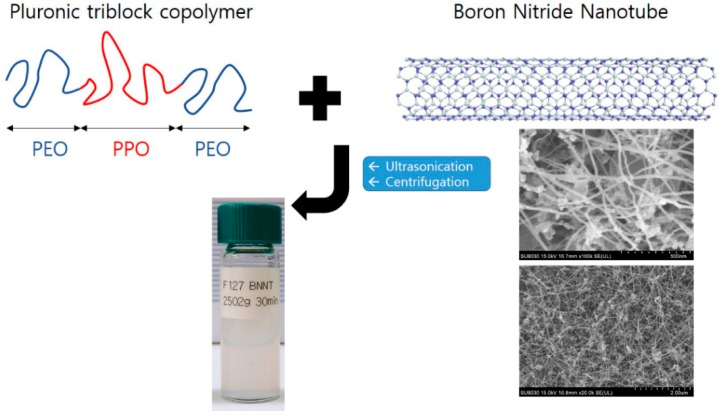
A schematic view of the sample preparation for Pluronic triblock copolymer-boron nitride nanotube (BNNT) dispersion in aqueous solution. The photo is the SEM image of unmodified BNNTs with two different magnifications. The diameter and the length of BNNT are ca. 20 nm and above 1 µm, respectively.

**Figure 2 polymers-11-00582-f002:**
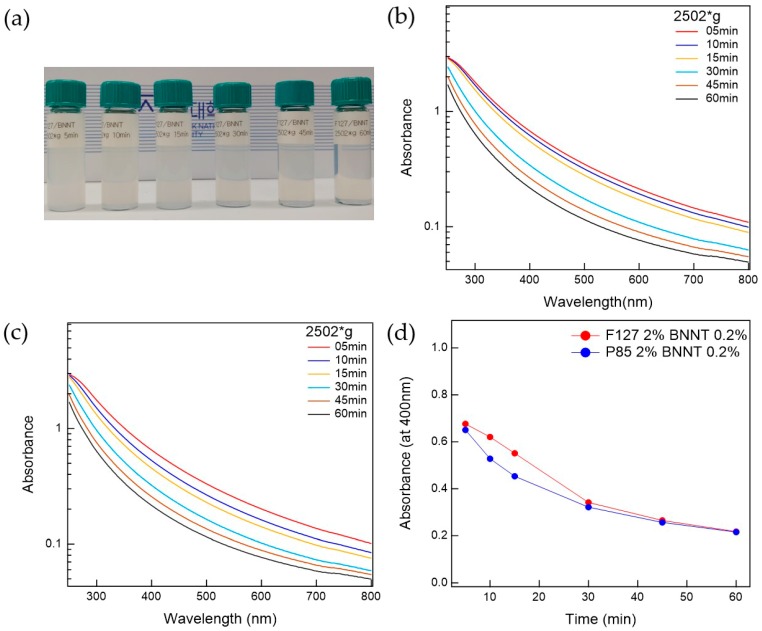
(**a**) Photo of F127-BNNT dispersions with different centrifuge times. UV–Vis measurements for (**b**) F127-BNNT and (**c**) P85-BNNT dispersion relative to centrifugation time. (**d**) Comparison between F127-BNNT and P85-BNNT dispersions’ absorbance at 400 nm versus centrifugation time.

**Figure 3 polymers-11-00582-f003:**
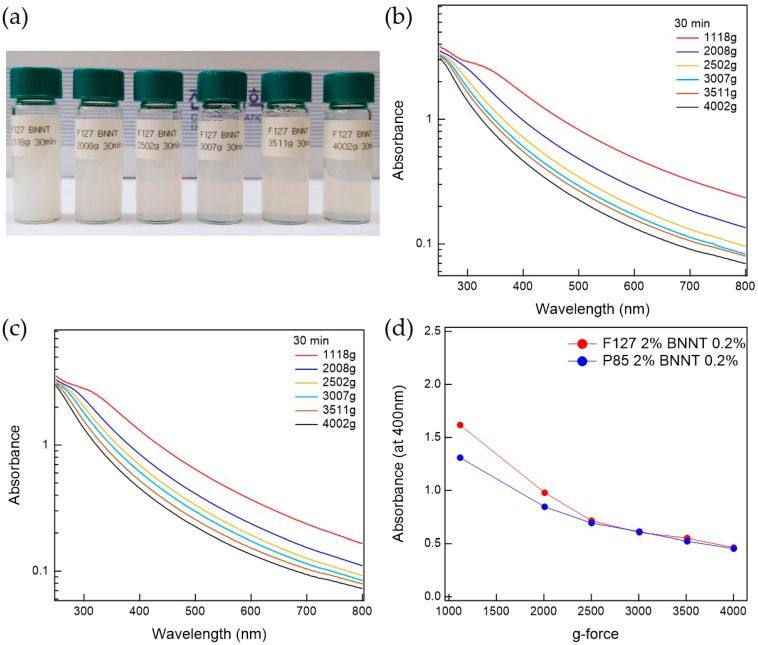
(**a**) Photo of F127-BNNT dispersions made at various g-forces. UV–Vis measurements for (**b**) F127-BNNT and (**c**) P85-BNNT dispersions depending on g-force. (**d**) Comparison between F127-BNNT and P85-BNNT dispersions’ absorbance at 400 nm versus g-force.

**Figure 4 polymers-11-00582-f004:**
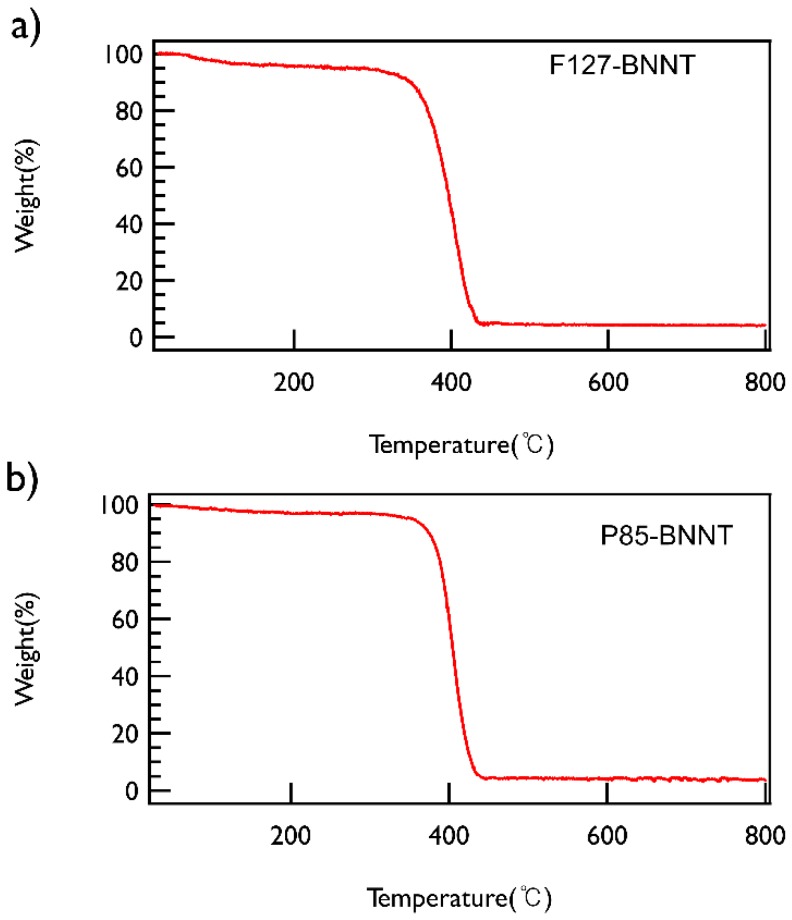
TGA data of (**a**) F127-BNNT and (**b**) P85-BNNT powders. Both powders were heated from ambient temperature to 800 °C at a heating rate of 5 °C/min under nitrogen.

**Figure 5 polymers-11-00582-f005:**
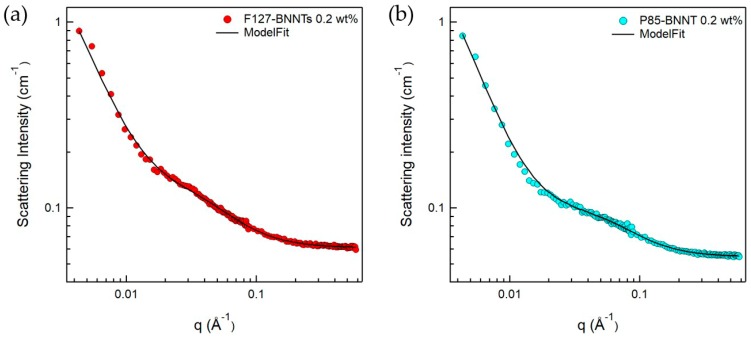
SANS intensities for (**a**) F127-BNNTs and (**b**) P85-BNNTs in D_2_O. The fitting model used is a sum model of the core-shell cylinder and Gaussian coil models.

**Figure 6 polymers-11-00582-f006:**
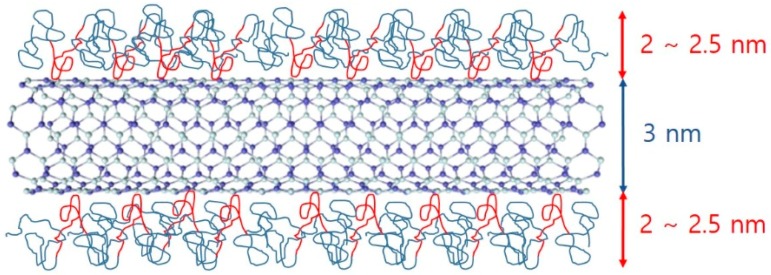
Schematic of polymer-BNNTs. Because of hydrophobic interactions between PPO and BNNTs, PPO parts (red) are attached to the surface of the BNNT and PEO parts (teal) contact water.

**Figure 7 polymers-11-00582-f007:**
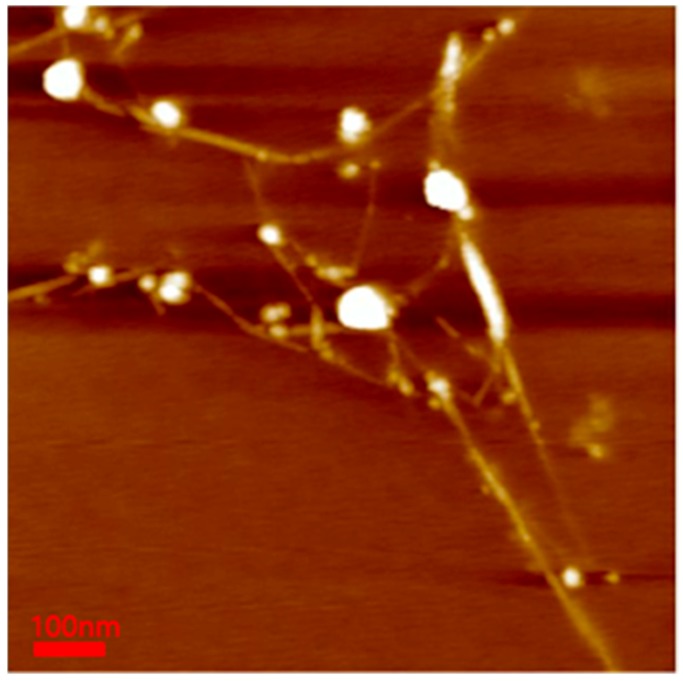
AFM image of the bare BNNT. The minimum diameter of BNNTs was 2.624 nm, and the maximum diameter was 7.072 nm.

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
