# Peer review of "Dispersion of Boron Nitride Nanotubes by Pluronic Triblock Copolymer in Aqueous Solution"

_polymers, 2019, doi:10.3390/polym11040582_

Round 1
Reviewer 1 Report
The paper entitled “Dispersion of Boron Nitride Nanotubes by Pluronic Triblock Copolymer in Aqueous Solution” the authors analysed the dispersion of boron nitride nanotubes (BNNTs) in aqueous solution, they utilized Pluronic F127 (PEO 99 PPO 69 PEO 99 , MW = ~ 12600) and Pluronic P85 (PEO 25 PPO 40 PEO 25 , MW = ~ 4600), specifically, Pluronic triblock copolymers are amphiphilic, having both hydrophilic Polyethylene oxide (PEO) group and relatively hydrophobic Polypropylene oxide (PPO) group as a dispersant.
The authors studied the characteristics of proposed samples by analysing the main properties.
The paper is interesting, I have no hesitation to suggest the publication, in addition the manuscript can be accepted in present form.
Reviewer 2 Report
The authors have made satisfactory changes and the manuscript is OK for publication
This manuscript is a resubmission of an earlier submission. The following is a list of the peer review reports and author responses from that submission.
Round 1
Reviewer 1 Report
The authors report the wrapping of BNNTs using Pluronic P85 and F127 polymers for better dispersion in water. The topic is of great scientific and technological interest. However, the study was not conducted in a comprehensive and convincing way. Many conclusions are given in the manuscript without experimental evidence. One example, L.192: “Considering the BNNT structure (the diameter (a few nanometers) and length (a few micrometers) of pristine BNNT, the bare BNNT is chopped during the sample preparation (upon ultrasonication and purification) and exists as an individual form, which is consistent with the SANS results.” There is no information on the diameter and length of the initial BNNTs without sonication at all, and it is very unprofessional to conclude on the diameter and length of BNNTs just based on one AFM images that containing a few nanotubes. Other problems:
L.113: “As the centrifugation time increases (at 2502 g-113 force), the UV-vis absorbance of Pluronic polymer-BNNT dispersion was decreased (i.e. the transparency of the dispersion was increased (Fig. 2a)) and then, it was gradually saturated after a specific centrifugation time (30 minutes) as shown in Fig. 2b, 2c and 2d” I couldn’t see any saturation and it is not reasonable to select 30mins.
I cannot comment much on the SANS results, but in general BNNT is a strong neutron absorber. How could this affect the SANS results?
Captions of Fig.2 and 3 are not correct.
There are many omissions and mistakes in writing. For example, L.102 “To separate the bundle BNNT in water, which can remain in the solution even ultrasonication” “after” is missing. Two lines below: “To get an optical condition for the preparation of BNNT dispersion in water” it should be “optimized”.
Reviewer 2 Report
The paper entitled “Dispersion of Boron Nitride Nanotubes by Pluronic Triblock Copolymer in Aqueous Solution” the authors described the dispersion of boron nitride nanotubes (BNNTs) in aqueous solution, they utilized Pluronic F127 (PEO 99 PPO 69 PEO 99 , MW = ~ 12600) and Pluronic P85 (PEO 25 PPO 40 PEO 25 , MW = ~ 4600), specifically, Pluronic triblock copolymers are amphiphilic, having both hydrophilic Polyethylene oxide (PEO) group and relatively hydrophobic Polypropylene oxide (PPO) group as a dispersant.
The hydrophobic surface of BNNTs was reformed into the hydrophilic surface by wrapping the polymer on the BNNT surface, making stable BNNT dispersion in aqueous solution.
The authors determined the dispersion characteristics (quality, stability and structure) of Pluronic polymer-BNNTs by UV-vis spectroscopy, thermogravimetric analysis (TGA) and small angle neutron scattering (SANS) and atomic for microscopy (AFM).
The authors used some different methodology to evaluate the properties of the proposed samples.
The paper is interesting, but some clarifications need to increase the value of the manuscript. But after minor revision I have no hesitation to suggest the publication.
Specific Comments:
-Abstract: The authors are invited to rewrite the abstract. Please avoid to start the abstract: “Due to their…..”
-Thermogravimetric Analysis (TGA) measurements –paragraph 2.4: The authors are invited to insert the conditions followed to perform the TGA analysis.
-Result and discussion: The authors are invited to insert the figure relative to the TGA graphs.